# Molecular Mechanisms of Succinimide Formation from Aspartic Acid Residues Catalyzed by Two Water Molecules in the Aqueous Phase

**DOI:** 10.3390/ijms22020509

**Published:** 2021-01-06

**Authors:** Tomoki Nakayoshi, Koichi Kato, Shuichi Fukuyoshi, Ohgi Takahashi, Eiji Kurimoto, Akifumi Oda

**Affiliations:** 1Graduate School of Pharmacy, Meijo University, 150 Yagotoyama, Tempaku-ku, Nagoya 468-8503, Aichi, Japan; 184331503@ccmailg.meijo-u.ac.jp (T.N.); kato-k@kinjo-u.ac.jp (K.K.); kurimoto@meijo-u.ac.jp (E.K.); 2Institute of Medical, Pharmaceutical and Health Sciences, Kanazawa University, Kakuma-machi, Kanazawa 920-1192, Ishikawa, Japan; fukuyosi@p.kanazawa-u.ac.jp; 3Department of Pharmacy, Kinjo Gakuin University, 2-1723 Omori, Moriyama-ku, Nagoya 463-8521, Aichi, Japan; 4Faculty of Pharmaceutical Sciences, Tohoku Medical and Pharmaceutical University, 4-4-1 Komatsushima, Aoba-ku, Sendai 981-8558, Miyagi, Japan; ohgi@tohoku-mpu.ac.jp; 5Institute for Protein Research, Osaka University, 3-2 Yamadaoka, Suita, Osaka 565-0871, Osaka, Japan

**Keywords:** d-amino acid, nonenzymatic reaction, reaction mechanism, age-related disease, quantum chemical calculation

## Abstract

Aspartic acid (Asp) residues are prone to nonenzymatic isomerization via a succinimide (Suc) intermediate. The formation of isomerized Asp residues is considered to be associated with various age-related diseases, such as cataracts and Alzheimer’s disease. In the present paper, we describe the reaction pathway of Suc residue formation from Asp residues catalyzed by two water molecules using the B3LYP/6-31+G(d,p) level of theory. Single-point energies were calculated using the MP2/6-311+G(d,p) level of theory. For these calculations, we used a model compound in which an Asp residue was capped with acetyl and methylamino groups on the *N*- and *C*-termini, respectively. In the aqueous phase, Suc residue formation from an Asp residue was roughly divided into three steps, namely, iminolization, cyclization, and dehydration, with the activation energy estimated to be 109 kJ mol^−1^. Some optimized geometries and reaction modes in the aqueous phase were observed that differed from those in the gas phase.

## 1. Introduction

Of the proteinogenic amino acids, aspartic acid (Asp) is relatively reactive, and Asp residues in peptides and proteins can be site-specifically isomerized [1,2,3,4], with nonenzymatic isomerization proceeding via a five-membered-ring succinimide (Suc) intermediate (Scheme 1) [1,2,3,4,5]. The Suc intermediate is formed via a nucleophilic attack of the main-chain amide nitrogen of the *C*-terminal-side adjacent residue ((*n* + 1) residue) on the side-chain carboxyl carbon of the Asp residue. Suc residues are prone to stereoinversion, with some l-Suc residues being converted to d-Suc residues. Subsequent hydrolysis of l-Suc residues forms l-Asp and l-isoAsp, and hydrolysis of d-Suc residues forms d-Asp and d-isoAsp. Typically, the hydrolysis of Suc residue produces Asp and isoAsp residues at a molar ratio of approximately 1:3 [4,6,7]. These isomerized Asp residues (i.e., l-isoAsp, d-Asp, and d-isoAsp) have been detected in various aging tissues, such as eye lenses [1,8,9,10,11], brains [12,13], skin [14], ligaments [15], aortas [16], teeth [17,18], and bones [19]. The formation of isomerized Asp residues disrupts the three-dimensional structures of peptides and proteins and is considered to be associated with various age-related diseases, such as cataracts [1,8,9,10,11] and Alzheimer’s disease [12,13]. Isomerized Asp residues are more abundant in the eye lenses in cataract patients than those of healthy individuals [11], and the isomerized Asp levels in the eye lenses tend to increase with aging [3,8,9,10,11]. A detailed analysis of αA- and αB- crystallin by Fujii et al. showed that Asp58 and Asp151 in αA-crystallin and Asp62 in αB-crystallin are highly isomerized [3,8,9]. Additionally, the levels of isomerized Asp residues are higher in water-insoluble fractions than in the water-soluble fractions at all sites of crystallin, and the formation of isomerized Asp residues is considered to contribute to its aggregation and insolubility [3].

It has been experimentally confirmed that Asp residue isomerization can occur in various buffers [4,5,20,21], and we have computationally demonstrated that water molecules [22,23,24,25,26,27], acetic acid [28], and dihydrogen phosphate ions [29,30] can catalyze Asp residue isomerization. Suc residue formation from Asp residues consists of two processes: cyclization via nucleophilic attack of the (*n* + 1) residue and dehydration of the *gem*-diol intermediate that is formed during cyclization (Scheme 2). The activation energy of the direct cyclization from the Asp residue to the *gem*-diol intermediate is high when the water molecules act as catalysts [22,31], which is assumed to arise from the weak nucleophilicity of the amide nitrogen. Therefore, we proposed a reaction mechanism in which cyclization occurs after the *C*-terminal peptide bond of the Asp residue is iminolized (Scheme 3). The iminol nitrogen with a high electron density due to the resonance effect can be expected to have enhanced nucleophilicity compared to the amide nitrogen. As a result, the activation energy was significantly reduced, which is consistent with experimentally determined activation energies [22,25,26,27]. However, in studies of water-catalyzed molecular mechanisms, calculations were conducted for the gas phase and did not include hydration effects. However, cytoplasmic proteins are constantly exposed to water, and the inclusion of hydration effects is important. In the present paper, we propose a molecular mechanism by which Suc is formed from Asp residues in the aqueous phase using a polarization continuum model (PCM).

## 2. Results and Discussion

Figure 1 presents the model compound used in the present study in which an Asp residue was capped with acetyl (Ace) and methylamino (Nme) groups on the *N*- and *C*-termini, respectively. Since nucleophilic attack by the main-chain nitrogen of the (*n* + 1) residue on the side-chain carboxyl carbon of the Asp residue is necessary for the formation of the Suc residue, it is preferable for the side-chain carbonyl group to be protonated (COOH) rather than deprotonated (COO^−^). Therefore, in our model compound, the side-chain carboxyl group was taken to be protonated. The dihedral angles *φ* (C–N–C_α_–C) and *ψ* (N–C_α_–C–N) characterized the main-chain conformation, and the dihedral angle *χ*_1_ (N–C_α_–C_β_–C_γ_) characterized the side-chain conformation. Conformational changes in which the dihedral angles changed by 30° or more were defined as “large conformational changes.”

All energy minima and transition state (TS) geometries were optimized using a B3LYP exchange-correlation functional with the 6-31+G(d,p) basis set, and single-point energy calculations were conducted for all optimized geometries using second-order Møller–Plesset perturbation theory (MP2) with the 6-311+G(d,p) basis set. All relative energies were calculated using MP2 single-point energies that were corrected for zero-point energies (ZPEs) and Gibbs energies (given at 1.00 atm at 298.15 K).

A reaction pathway from the reactant complex AM to the product complex SI was identified and the calculated energy profile of this reaction pathway is presented in Figure 2. In this figure, the amide-form Asp residue with a deprotonated side-chain carboxyl group, iminol tautomer Asp residue, and *gem*-diol tetrahedral intermediate are abbreviated as DP, IM, and TH, respectively. Six TSs are numbered consecutively as TS1–TS6.

### 2.1. Cyclization Step

The reactant complex AM was a complex consisting of an amide-form Asp residue and two water molecules W1 and W2, where the two water molecules were placed around the Asp residue and connected to the main-chain carbonyl carbon and side-chain carboxyl OH proton via a hydrogen bond network (Figure 3a). Unlike previous studies in the gas phase [22], no hydrogen bond was formed between the amide NH proton of the (*n* + 1) residue and the side-chain carboxyl oxygen of the Asp residue in the aqueous phase. In addition, a hydrogen bond was formed between W1 and the main-chain carbonyl oxygen of the *N*-terminal adjacent residue ((*n* − 1) residue) (1.876 Å).

The iminolization process, which started from AM, consisted of two steps. First, AM was converted to the DP via TS1. Second, the DP was converted to IM1 via TS2. The relative energies of the two TSs TS1 and TS2 relative to AM were 41.1 and 51.4 kJ mol^−1^, respectively. The optimized geometries of TS1, DP, TS2, and IM1 are presented in Figure 3b–e, respectively. During the first step of the iminolization process, the double-proton transfer from the side-chain carboxyl group of the Asp residue to W1 occurred, which was mediated by W2. In this step, the side-chain carboxyl group of the Asp residue was deprotonated and W1 was protonated, with the resulting formation of the DP consisting of an amide-form Asp residue with a deprotonated side-chain carboxyl group, a hydronium cation W1, and a water molecule W2, as presented in Figure 3b. Both the hydrogen bonds connecting W1 and the carbonyl oxygen of the (*n* − 1) residue and those connecting W1 and the carbonyl oxygen of the Asp residue of the DP were much shorter than those of AM and TS1. These hydrogen-bond enhancements may have contributed to the stabilization of the DP. In addition, previous studies on a two-water-catalyzed pathway in the gas phase did not identify an optimized geometry for the DP, and iminolization proceeded via a single step [22]. Therefore, the hydration effects may have also been important for stabilizing the DP. Subsequently, proton donation from the hydronium cation W1 to the main-chain carbonyl oxygen of the Asp residue and proton abstraction from the amide nitrogen of the (*n* + 1) residue via a side-chain carboxyl oxygen occurred, and the DP was converted to IM1 via TS2. During the conversion from the DP to TS2, the C_β_–C_γ_ bond rotated, a hydrogen bond was formed between the main-chain amide NH proton and side-chain carboxyl oxygen, and proton transfer occurred. This resulted in side-chain carboxyl group reprotonation.

The iminolization process consisted of two steps and proceeded via a quadruple proton transfer that was mediated by two water molecules and a side-chain carboxyl group. This quadruple proton transfer occurred in a stepwise fashion rather than a concerted fashion. During the conversion from AM to IM1, the dihedral angles *φ*, *ψ*, and *χ*_1_ changed by less than 20°; however, a large C_β_–C_γ_ bond rotation occurred during the second iminolization step. Therefore, the flexibility of the side chain was considered important for iminolization progression in the aqueous phase. 

As shown in Figure 2, the energy of the DP, after correction for the ZPE and Gibbs energy, was 0.973 kJ mol^−1^ higher than that of TS1. However, intrinsic reaction coordinate (IRC) calculations and the subsequent geometry optimization confirmed that the DP was an energy-minimum geometry directly connected to TS1 and that the energy of the DP calculated at the B3LYP/6-31+G(d,p) level of theory before energy corrections related to the ZPE and Gibbs energy was 3.44 kJ mol^−1^ lower than that of TS1. These computational results indicate that the DP was located at a very shallow stationary point on the potential energy surface. 

### 2.2. Conformational Changes of the Iminol Tautomer

In IM1, the product of the iminolization process, a hydrogen bond was formed between the main-chain iminol nitrogen and the side-chain carboxyl OH proton (see Figure 3e), where this hydrogen bond interfered with the direct cyclization from IM1. To form IM3, which contains a favorable conformation for cyclization, a two-step conformational change from IM1 was required. IM1 was first converted to IM2 via TS3 with a local activation energy of 10.3 kJ mol^−1^ in the first step, followed by the conversion of IM2 to IM3 via TS4 with a local activation energy of 31.6 kJ mol^−1^ in the second step. The optimized geometries of TS3, IM2, TS4, and IM3 are presented in Appendix A. In the first step, the hydrogen bond between the main-chain iminol nitrogen and the side-chain carboxyl OH proton was completely cleaved via rotation of the C_β_–C_γ_ bond. In the second step, a *cis*-*trans* rotation of the side-chain carboxyl group occurred, converting the *trans*-carboxyl group to a *cis*-carboxyl group. The IM3 formed in this step was 4.79 and 12.3 kJ mol^−1^ more stable than IM1 and IM2, respectively. During these conformational changes of the iminol tautomer, all intermolecular hydrogen bonds formed between the water molecules and iminol tautomer Asp residue were maintained.

### 2.3. Cyclization Process

The cyclization process starting from IM3 proceeded via TS5 with a local activation energy of 62.9 kJ mol^−1^, resulting in the formation of TH1. The optimized geometries of TS5 and TH1 are presented in Figure 4. The energies of TS5 and TH1 relative to AM were 105 and 7.35 kJ mol^−1^, respectively. In this process, the main-chain iminol nitrogen nucleophilically attacked the side-chain carboxyl carbon. The distances between the main-chain iminol nitrogen and the side-chain carboxyl carbon were 3.239, 1.598, and 1.479 Å for IM3, TS5, and TH1, respectively, and a covalent bond between the iminol nitrogen and the carboxyl carbon was formed during this process. In addition, a triple proton transfer from the main-chain iminol OH group to the side-chain carboxyl oxygen occurred, which was mediated by two water molecules. During the conversion from IM3 to TH1, the dihedral angles *ψ* and *χ*_1_ changed by 41° and 31°, respectively. Therefore, the flexibilities of the main and side chains are suggested to be required for iminolization progression. Another potential reaction pathway was found in which a *gem*-diol intermediate was directly formed from IM2 (i.e., cyclization occurred before IM2 underwent a *cis*-*trans* rotation); however, the energy of the TS relative to AM of this reaction pathway was estimated to be 115 kJ mol^−1^, which was 10.5 kJ mol^−1^ higher than that of TS5.

### 2.4. Dehydration Process

A direct dehydration pathway from TH1 to form the Suc residue with a low activation energy could not be determined. In TH1, two OH groups in the *gem*-diol moiety were not connected by two catalytic water molecules, with direct dehydration of TH1 requiring a proton abstraction from one OH group by an oxygen of the other OH group. The TS of the direct dehydration pathway for the formation of Suc from TH1 is presented in Appendix A, with the energy of the corresponding TS relative to AM being 151 kJ mol^−1^. As presented in Appendix A, the corresponding TS included an extremely strained four-membered ring, which may have contributed to the high activation energy of this reaction. 

A dehydration pathway with a low activation energy could be started from TH2. TH2 and TH1 are both complexes consisting of a *gem*-diol intermediate and two water molecules; however, the arrangement of water molecules in TH2 was substantially different from that in TH1. In TH2, two water molecules connected two OH groups in the *gem*-diol moiety via a hydrogen bond network. Dehydration starting from TH2 to form the product complex SI proceeded via TS6 with a local activation energy of 90.9 kJ mol^−1^. The optimized geometries of TH2, TS6, and SI are presented in Figure 5. During dehydration, a proton relay occurred from one OH group to the other in the *gem*-diol moiety mediated by two water molecules along the hydrogen bond network. In addition, one of the C–O bonds was cleaved in concert with the triple proton transfer, and a water molecule was released. The energies of TH2, TS6, and SI relative to AM were 17.9, 109, and −49.5 kJ mol^−1^, respectively. During the conversion of TH2 to SI, all changes in the defined dihedral angles *φ*, *ψ*, and *χ*_1_ were less than 20° and dehydration proceeded without significant conformational changes of the main and side chains. As presented in Figure 2, the energy of TS6 was higher than that of other TSs; however, there was no substantial difference between the energies of TS5 and TS6. In contrast, the energies of TS5 and TS6 were much higher than those of other TSs. Thus, cyclization and/or dehydration processes were likely the rate-determining steps of the entire reaction pathway from AM to SI.

## 3. Materials and Methods

As in previous studies [22,23,26,27,28,30], the model compound Ace–Asp–Nme was used (Figure 1). All calculations were conducted using Gaussian16 software [32]. The geometries of the energy minima and TSs were optimized without any constraints via density-functional theory (DFT) calculations using the B3LYP/6-31+G(d,p) level of the theory. Vibrational frequency calculations were conducted to confirm them as energy minima (with no imaginary frequencies) or TSs (with a single imaginary frequency). IRC calculations were performed for all TSs. If the IRC calculations were successful, the following full geometry optimization identified two energy minima directly connected to TS. In contrast, the IRC calculation for TS3 failed in the initial stage. Thus for TS3, full geometry optimization was conducted after all atoms were slightly displaced along the transition vector, and two energy minima directly connected to TS3 were identified. Furthermore, single-point calculations performed using the MP2/6-311+G(d,p) level of theory for all optimized geometries were used to calculate more reliable energies. The relative energies of all optimized geometries calculated at the MP2/6-311+G(d,p) level of theory were corrected using ZPEs and the thermodynamic corrections (to give the Gibbs energies at 1.00 atm and 298.15 K) were calculated at the B3LYP/6-31+G(d,p) level of theory. To reproduce the aqueous conditions, PCM was employed for all calculations, and the dielectric constant of water was set to 78.355 (the default setting in Gaussian16).

## 4. Conclusions

The present study explored the molecular mechanism of the formation of Suc from an Asp residue catalyzed by two water molecules in the aqueous phase. The reaction pathway was roughly divided into three processes: iminolization, cyclization, and dehydration. The proton relay in the iminolization process was mediated by the side-chain carboxyl group and two catalytic water molecules, whereas the proton relays in the cyclization and dehydration processes were mediated by two water molecules. Two-water-catalyzed iminolization processes in the aqueous phase proceeded in a stepwise fashion, unlike for the gas phase. During the iminolization process, a large rotation of the side-chain C_β_–C_γ_ bond occurred, and large conformational changes of the main and side chains were also observed. This suggests that the flexibilities of both the main and side chains are important for Asp residue isomerization. Recently, we conducted molecular dynamics simulations for elastin and αA-crystallin peptides in which the rate of Asp-residue isomerization was experimentally determined and showed that the flexibility of the sites around the Asp residues is important for the Asp-residue isomerization [33,34]. These reports support the experimental and computational results from the present study.

The activation energies of cyclization (105 kJ mol^−1^) and dehydration (109 kJ mol^−1^) were higher than that of iminolization (51.4 kJ mol^−1^), and it was assumed that cyclization and/or dehydration processes were the rate-determining steps. Previously, the activation energy of Asp-residue isomerization in distilled water was experimentally estimated to be 25.7–29.0 kcal mol^−1^ (108–121 kJ mol^−1^), and the activation energy calculated in the present study was consistent with this experimental value. In this study, we introduced hydration effects and the Gibbs energies, which were not included in previous studies, and evaluated the relative energies using more reliable MP2 single-point calculations. However, there was no substantial difference between the activation energy reported in the present paper and in previous studies. Therefore, it appears that the corrections introduced in this study did not significantly affect the activation energy of Suc residue formation from an Asp residue. In contrast, the optimized geometries and reaction modes in the present study exhibited some differences from those in the gas phase. Cytoplasmic proteins are always exposed to water, and solvents affect the chemical properties of solutes; thus, the corrections introduced in this study are considered to be important for accurately predicting the reaction pathway in the aqueous phase.

## Data Availability

Data is contained within the Article and Appendix A.

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
