# Peer review of "Molecular Mechanisms of Succinimide Formation from Aspartic Acid Residues Catalyzed by Two Water Molecules in the Aqueous Phase"

_ijms, 2021, doi:10.3390/ijms22020509_

Round 1
Reviewer 1 Report
The manuscript entitled “Molecular Mechanisms of Succinimide Formation from Aspartic Acid Residues Catalyzed by Two Water Molecules in the Aqueous Phase” deals with the proposal of a molecular mechanism by which succinimide is formed from aspartic acid residues in aqueous phase using a polarization continuum model via an iminol tautomer and a gem-diol intermediate.
The simulations used in this manuscript are adequate for its goal. The results obtained are clearly presented.
The manuscript is well written and easy to understand. The references used in the manuscript are recent and are adequate. Regarding the novelty of the manuscript as far as I am concerned this is the first time that hydration effects and Gibbs energies were included, and used more reliable MP2 single point calculations, which provided the subsequent alterations in the optimized geometries of the intermediates.
In my opinion, the findings shown here are interesting for a broader community and deserve to be published.
Author Response
We thank the reviewer 1 for the valuable comment. We are deeply grateful for the time and energy the reviewer spent.
Reviewer 2 Report
This manuscript describes quantum chemical calculations of a possible mechanism for the title conversion reaction. The paper begins with a well written introduction that explains the issue and presents the current status of the problem. The authors next coherently lay out their plan for examining this reaction, and place the system within a medium that is designed to simulate aqueous solution, although they do include several discrete water molecules which are integral to the purported mechanism. The calculations are carried out at a suitable level for a problem of this nature, and the authors do an excellent job of explaining the entire process, with well thought out figures that compactly describe the salient issues of each reaction step. It is gratifying that the authors present their energetics in free energy terms instead of the electronic energy which is so common in the literature. The text is concisely written, with an eye on the ball at all times, maintaining focus on the most important aspects without unnecessary tangents. Although the study of a proposed mechanism is never able to prove definitively to be the correct one, this paper lays out a compelling case. Consequently, the manuscript can be recommended for publication with some enthusiasm.
The only suggestion would be for some added labeling of the figures to make it easier for a reader to identify specific intermediates and transition states. For example, Fig 3 might add the labels AM, TS1, DP etc right next to each structure, with similar requests for the other figures.
Author Response
We thank the reviewer 2 for the helpful comment. We are deeply grateful for the time and energy the reviewer spent. We have revised the manuscript following the reviewer’s comment.
# Reviewer 2
The only suggestion would be for some added labeling of the figures to make it easier for a reader to identify specific intermediates and transition states. For example, Fig 3 might add the labels AM, TS1, DP etc right next to each structure, with similar requests for the other figures.
-> We have added the labels to identify each optimized geometry to the Figures 3, 4, and 5.